# Association of RNA-modification "writer" genes with prognosis and response to immunotherapy in patients with low-grade glioma

Lupeng Zhang[1⊙], Chiwen Qu[2,3⊙], Chen Shi[1⊙], Fan Wu[1⊙], Yifan Tang[4], Yue Li[5], Jinlong Li[1], Huicong Feng[4], Suye Zhong[4], Jun Yang[4], Xiaomin Zeng[6]*, Xiaoning Peng[1,2,4]*

1 Department of Biochemistry and Molecular Biology, Jishou University School of Medicine, Jishou, Hunan, China, 2 Department of Statistics, College of Mathematics and Computer Science, Hunan Normal University, Changsha, Hunan, China, 3 School of Public Health and Management, Youjiang Medical University for Nationalities, Baise, Guangxi, China, 4 Department of Pathology and Pathophysiology, Hunan Normal University School of Medicine, Changsha, Hunan, China, 5 Department of Pathology, Xiangtan Central Hospital, Xiangtan, China, 6 Department of Epidemiology and Health Statistics, Xiangya Public Health School, Central South University, Changsha, Hunan, China

⊙ These authors contributed equally to this work.
* pxiaoning@hunnu.edu.cn (XP); zxiaomin@csu.edu.cn (XZ)

**Data Availability Statement:** The datasets generated and/or analyzed during the current study are available in the public open platform, including CGGA (http://www.cgga.org.cn/) and TCGA

## Abstract

RNA modification is a key regulatory mechanism involved in tumorigenesis, tumor progression, and the immune response. However, the potential role of RNA modification "writer" genes in the immune microenvironment of gliomas and their effect on the response to immunotherapy remains unclear. The purpose of this study was to evaluate the role of RNA modification "writer" gene in the prognosis and immunotherapy response of low-grade glioma (LGG). The consensus non-negative matrix factorization (CNMF) method was used to identify different RNA modification subtypes. We used a novel eigengene screening method, the variable neighborhood learning Harris Hawks optimizer (VNLHHO), to screen for eigengenes among the RNA modification subtypes. We constructed a principal components analysis score(PCA_score)-based prognostic prediction model and validated it using an independent cohort. We also analyzed the association between PCA_score and the immune and molecular features of LGG. The results suggested that LGG can be divided into two different RNA modification-based subtypes with distinct prognostic and molecular features. High PCA_score was significantly associated with a poor prognosis in LGG and was an independent prognostic factor. A nomogram containing PCA_score and clinical features was constructed, and it showed a significant predictive value. PCA_score was negatively correlated with tumor purity and the abundance of CD4+ T cells in LGG patients. LGG patients with high PCA_score had lower Tumor Immune Dysfunction and Exclusion scores and showed an immunotherapy response. In conclusion, we report a novel RNA modification-based prognostic model for LGG that lays the foundation for evaluating LGG prognosis and developing more effective therapeutic strategies for these tumors.

(https://portal.gdc.cancer.gov/). The CGGA-LGG data: "Home" - "Download"; the TCGA-LGG clinical data set: "Project"-"TCGA-LGG", "Data Category"-"clinical"; theTCGA-LGG RNA-seq: "Project"-"TCGA-LGG", "Data Category"-"transcriptome profiling", "Data Type"-"Gene Expression Quantification".

**Funding:** This project was supported by the National Natural Science Foundation of China (grant number 81472860, 61761001), Key R & D project of Hunan Province (grant number 2020DK2002) and the research start-up fund for Prof. Peng Xiaoning from Jishou University (grant number 91602-111900). The funders had no role in study design, data collection and analysis, decision to publish, or preparation of the manuscript.

**Competing interests:** The authors have declared that no competing interests exist.

**Abbreviations:** WHO, world health organization; LGG, low-grade gliomas; VNLHHO, variable neighborhood learning Harris Hawks optimizer; CGGA, chinese glioma genome atlas; TCGA, the cancer genome atlas; TME, tumor microenvironment; OS, overall survival; RPKM, reads per kilobase transcriptome per million reads; FPKM, Fragments Per Kilobase Million, Fragments Per Kilobase of exon model per Million mapped fragments; TPM, Transcripts per million, Transcripts Per Kilobase of exon model per Million mapped reads; PCA, principal components analysis; C-index, concordance index; ROC, receiver operating characteristic; DCA, decision curve analysis; CNVs, copy number variations; GSVA, gene set variation analysis; DEG, Differentially expressed gene; CNMF, consensus non-negative matrix factorization; GO, gene ontology; KEGG, kyoto encyclopedia of genes and genomes; ICB, Immune checkpoint blockade; TIDE, Tumor Immune Dysfunction and Exclusion.

# Introduction

Gliomas originate from glial cells and are common malignant tumors of the central nervous system [1]. The annual incidence rate of gliomas is approximately 6 per 100,000 individuals worldwide [2]. In 2016, the World Health Organization (WHO) reclassified tumors of the central nervous system. Grade II and III gliomas were combined into one category, low-grade glioma (LGG), and grade IV gliomas were designated as glioblastomas (GBMs) [3].

LGG is a heterogeneous disease with a distinct pathogenesis and involves somatic mutations, genetic fusions, genetic instability, and epigenetic alterations [4–6]. Postoperative radiotherapy and chemotherapy are the main treatment options for LGG. However, due to disease heterogeneity and drug resistance, the prognosis of LGG has not improved significantly [7]. For example, temozolomide (TMZ) is currently the main chemotherapy drug for glioma, which plays a therapeutic role through non selective DNA damage. Under this treatment, almost all glioma patients have tumor recurrence,and the recurrent tumors will usually have significant drug resistance, which leads to poor prognosis of glioma patients [8]. Therefore, there is an urgent need to identify new prognostic factors and improve the treatment of LGG.

Evidence shows that RNA modification is an important epigenetic mechanism and is involved in physiological changes and disease development [9, 10]. RNA modification is also known to play an important role in tumor genomics. The most frequent RNA modifications are the methylation of the sixth and first nitrogen atoms ($m^6A$ and $m^1a$) of adenine. In addition, alternative Poly(A) (APA) and RNA editing (A-to-I) also occur. These modifications mainly regulate gene expression and influence the tertiary structure of ribosomes, thereby regulating cellular self-renewal, differentiation, and apoptosis and promoting the occurrence and progression of tumors [11–16]. These modifications are usually related to the activity of enzymes called "writer". Furthermore, there is a strong link between RNA modification and its associated enzymes and the tumor microenvironment (TME) [17–19].

Gene expression profiling has played a significant role in the identification and classification of tumor associated molecules. However, only few feature genes are closely involved with tumors. We previously proposed a new tumor signature gene acquisition method called the variable neighborhood learning Harris Hawks optimizer (VNLHHO), which can more effectively identify signature genes from tumor gene expression profiles [20].

In this study, we analyzed genomic data from the Chinese Glioma Genome Atlas (CGGA) and The Cancer Genome Atlas (TCGA) glioma patients to assess the patterns of RNA modification. We found that LGG can be classified into distinct RNA modification-based subtypes that are characterized by the infiltration of various immune cells. By further screening the differentially expressed RNA modification subtype genes using VNLHHO, we identified the characteristic genes related to RNA modification in LGG and constructed a PCA_score model. We demonstrated the effectiveness of PCA_score in LGG prognostication, highlighting its potential value in guiding immunotherapy.

# Materials and methods

## Ethics approval and consent to participate

Not applicable. This study only conducts data mining on public databases and does not involve any experiments. All methods were performed in accordance with the relevant guidelines and regulations.

## LGG dataset and preprocessing

Gene expression data on LGG and the clinical information were obtained from the Chinese Glioma Genome Atlas (CGGA) database and The Cancer Genome Atlas (TCGA) database

[21, 22]. Data on somatic mutations and somatic copy number variations (CNVs) were obtained from the TCGA database. Samples for which complete clinical information was unavailable were excluded from the study. Using the "SVA" software package, batch effect correction was performed on the two datasets obtained from the CGGA database [23]. Three formalin-fixed paraffin-embedded tumor samples were excluded from the TCGA dataset. A total of 974 LGG samples with detailed clinical data were collected, of which 474 samples obtained from the CGGA database were used as the training set and 500 samples obtained from the TCGA database were used as the validation set.

## Cluster typing and gene set variation analysis (GSVA) of 26 RNA modification "writer" enzymes

Information on 26 RNA modification "writer" enzymes including seven $m^6A$ modification enzymes, four $m^1A$ modification enzymes, 12 APA modification enzymes, and 3 A-I modification enzymes—was collected from previously published articles [16, 24, 25]. Then, LGG patients from the CGGA and TCGA datasets were divided into subgroups based on the CNMF method using the R package "CancerSubtypes" [26]. In order to examine differences in the biological functions of different RNA modification subtypes, the hallmark gene set (c2.cp.kegg.v7.2) in the MsigDB database was used for GSVA enrichment analysis [27].

## Analysis of the correlation between LGG molecular subtypes and clinical features and prognosis

In order to test the clinical significance of the different RNA modification-based molecular subtypes identified during CNMF clustering, clinicopathological characteristics were compared between different RNA modification-based molecular subtypes. The clinical characteristics included age, gender, grade, Chemo_status, Radio_status, *IDH*_status, 1p19q_status, and *MGMT*p_status. In addition, Kaplan-Meier survival analysis was performed using the R package "survival" to evaluate differences in overall survival (OS) between different RNA modification-based tumor subtypes.

## Analysis of the correlation between LGG molecular subtypes and the TME

The ESTIMATE algorithm was used to evaluate the Immune score, Stromal score, and tumor purity in each case [28]. In addition, the CIBERSORT algorithm was used to calculate the scores for 22 human immune cell subgroups in each LGG sample [29].

## Differentially expressed gene (DEG) identification, functional annotation and VNLHHO

Using the "limma" package in R [30], DEGs were identified after comparing tumor subtypes with different RNA modification "writer" expression (criteria: $|\log2FC| \geq 0.5$ and $p < 0.05$). To further understand the potential functions of DEGs associated with RNA modification "writer" subtypes and to identify the associated gene functions and enrichment pathways, functional enrichment analysis of the DEG was performed using the R package "clusterprofile". In addition, the VNLHHO algorithm was used to further determine the characteristic differentially expressed genes in order to identify the key genes involved in LGG (VNLHO is detailed in reference [19] and S1 File).

## Construction of the PCA_score based on RNA modification "writer" enzymes

An RNA modification scoring model was used to quantify the RNA modification pattern in individual tumors. This was called PCA_score. Specific steps were as follows:

1. Kaplan–Meier and univariate Cox regression analysis were used to screen for prognostic DEGs ($p < 0.001$).

2. DEGs associated with prognostic differences that also appeared on the list of characteristic genes were selected.

3. Principal component analysis (PCA) was used to obtain the score (PCA_score) of characteristic genes related to RNA modification. PCA1 and PCA2 were chosen as feature scores. A method similar to GGI was used to define PCA_score as follows [31, 32]:

$$\text{PCA\_score} = \sum (PC1_i + PC2_i)$$

where I is the expression of the related genes used to obtain PCA_score.

LGG patients were divided into low and high PCA_score groups using the "survminer" software package. The generated Kaplan–Meier survival curves and receiver operating characteristic (ROC) curves were used to evaluate the impact of PCA_score on of LGG prognosis.

## Construction of a PCA_score prognostic model

Kruskal-Wallis and Mann-Whitney tests were used to analyze the correlation between PCA_score and different clinical characteristics (gender, age, grade, Chemo_status, Radio_status, *IDH*_status, 1p19q_status, and *MGMT*p_status). Multivariate Cox analysis was adopted to determine whether PCA_score is an independent prognostic factor for LGG. The sensitivity of LGG to radiotherapy and chemotherapy was also compared between the low and high PCA_score groups.

Based on the results of independent prognostic analysis, the R package"rms" was used to build a predictive prognostic nomogram. In the nomogram, each variable was matched with a score, and the scores of all variables for each sample were added to obtain the total score [33]. ROC curves, the C-index, and calibration curves were used to evaluate the prediction efficiency of the model. The clinical utility of the predictive model was determined using decision curve analysis (DCA).

## Analysis of the correlation of PCA_score with the immune microenvironment and immune checkpoints

In order to evaluate the relationship between the TME and PCA_score, Pearson correlation coefficients were used and the correlation of PCA_score with the abundance of immune cell infiltration, immune score, stromal score, and tumor purity was determined. The correlation between PCA_score and the expression of immune checkpoint genes was also examined.

## Immunotherapy response prediction

Tumor Immune Dysfunction and Exclusion(TIDE) is a computational method that combines the expression characteristics of T cell dysfunction and rejection to simulate tumor immune evasion [34]. Based on preprocessing genomic data, a TIDE score was obtained to predict the clinical response of LGG to immune checkpoint blockade (ICB).

### Analysis of the correlation of PCA_score with LGG somatic mutations and drug sensitivity

In order to understand the relationship of PCA_score with LGG somatic mutations and the sensitivity to common chemotherapy drugs, the R package "maftools" was used. Genes with a mutation frequency of ≥5% in the H-PCA_score and L-PCA_score groups in the TCGA-LGG cohort were identified, and the tumor mutation burden (TMB) was calculated for each sample. Then, the R package "pRRophetic" was used to calculate the 50% inhibitory concentration (IC50) value of commonly used chemotherapy drugs [35]. The difference in gene mutation frequency and IC50 values between the H-PCA_score and L-PCA_score groups was determined using the chi-square test and the rank sum test, respectively.

### Analysis of the correlation between PCA_score and prognosis in the immunotherapy cohort

Groups of patients with advanced urothelial cancer (IMvigor210 cohort) and patients from the BLCA cohort who were treated using an anti-PD-L1 antibody (atezolizumab) were included. Transcriptome data and clinical information for the IMvigor210 cohort were obtained from the R package "IMvigor210CoreBiologies" [36]. Meanwhile, gene expression data and clinical information for the BLCA cohort was obtained from the TCGA website. The R package "DEseq2" was used to normalize the data and obtain TPM values.

### Statistical analyses

All statistical analyses were performed using R version 4.0.1. Further, $p < 0.05$ was considered statistically significant.

## Results

### Genetic and transcriptional changes in RNA modification "writer" enzymes in LGG

A total of 26 RNA modification "writer" genes were included in this study (S1 Table in S2 File). First, somatic mutation analysis were performed on these 26 RNA modification "writer" genes. The mutation frequency in the TCGA-LGG cohort was very low, and only 3.17% of samples showed mutations (Fig 1A). Examinations of somatic copy number changes revealed the presence of copy number changes in most RNA modification "writer" genes. The copy numbers in *CPSF4*, *ADARB2*, *KIAA1429*, *CSTF1*, and *TRMT6* were generally increased, while those in *ADARB2*, *CSTF3*, *RBM15*, *CPSF2*, *ZC3H13*, and *TRMT61A* were decreased (Fig 1B). The chromosomal positions of these CNV changes were also analyzed (Fig 1C). Expression differences in these RNA modification "writer" genes were compared between LGG tissues and normal tissues using the GEPIA database. Significant expression level differences were identified for most RNA modification "writer" genes (S1 Fig). Correlation analysis revealed extensive interactions between the expression levels of different RNA modification "writer" genes (Fig 1D). these results indicated that the interaction of RNA modification "writer" genes together mediate the regulation of RNA modification pathways during tumor progression.

### Identification of LGG subtypes based on RNA modification "writer" genes

In order to fully understand the potential role of RNA modification in the occurrence of LGG, data from 474 patients in the CGGA database (model cohort) and 500 patients in the TCGA

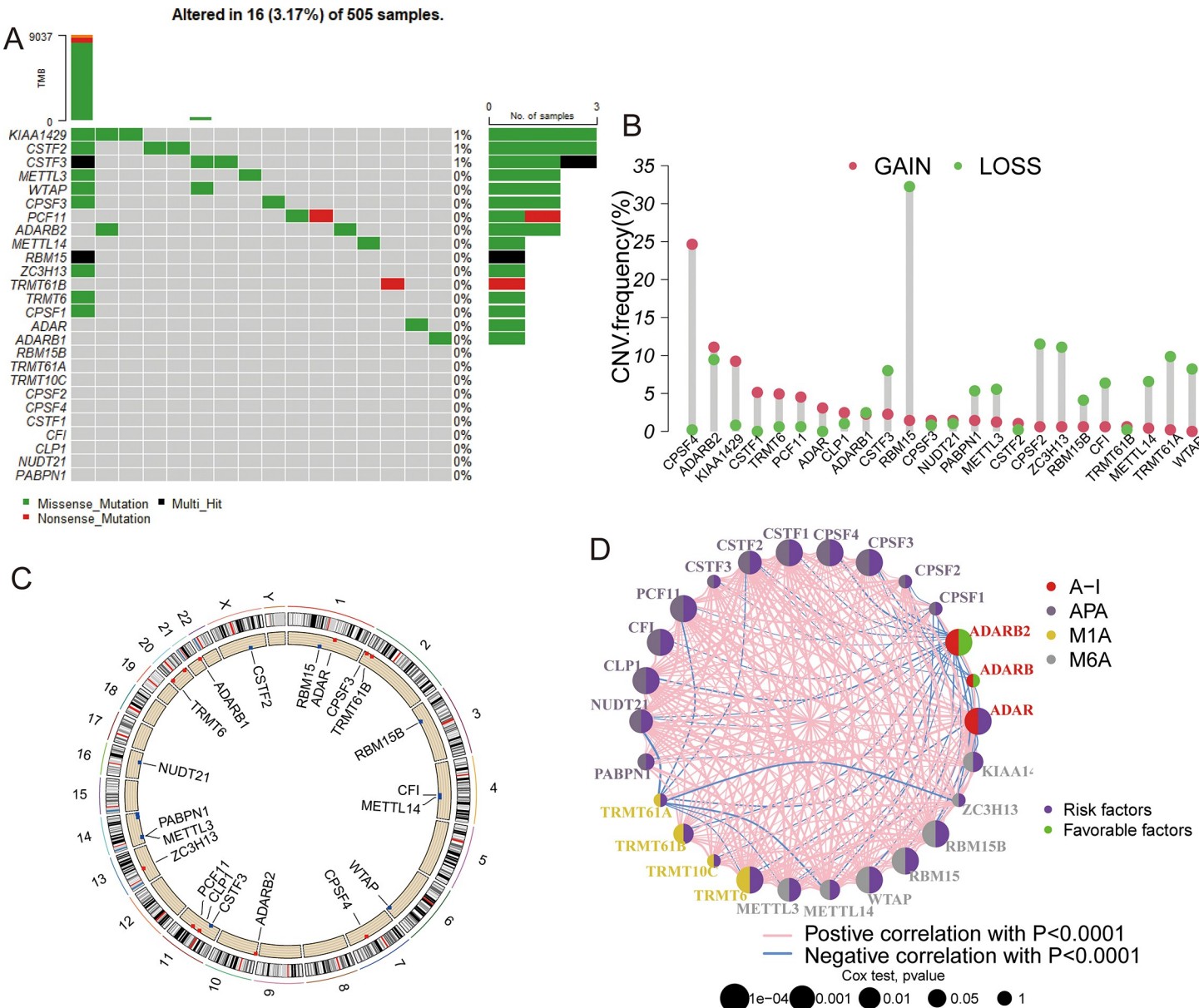

**Fig 1. Genetic alterations of RNA modification "writer" genes in LGG.** (A) Mutation frequencies of 26 RNA modification "writer" genes in 500 patients with LGG, from the TCGA cohort. (B) Frequencies of CNV gain, loss, and non-CNV among RNA modification "writer" genes. (C) Locations of CNV alterations in PRGs on 23 chromosomes. (D) Interactions between RNA modification "writer" genes in LGG. Lines connecting genes represent their interactions, and the thickness of the lines represents the strength of the association between genes.

database (validation cohort) were analyzed. Univariate Cox analysis revealed the prognostic value of the 26 RNA modification "writer" genes in LGG (S2 Table in S2 File).

CNMF was used to classify LGG patients. LGG patients from the model and validation cohorts could be divided into two subgroups. Different subgroups showed different survival outcomes, and there were clear boundaries between colored areas. The average contour width (ASW), and indicator of clustering consistency, can be used evaluate similarity between samples within a particular subgroup. The ASW values in the model group and validation group was 0.88 and 0.84, respectively (Fig 2A and S2A Fig). Therefore, the molecular subtypes (C1

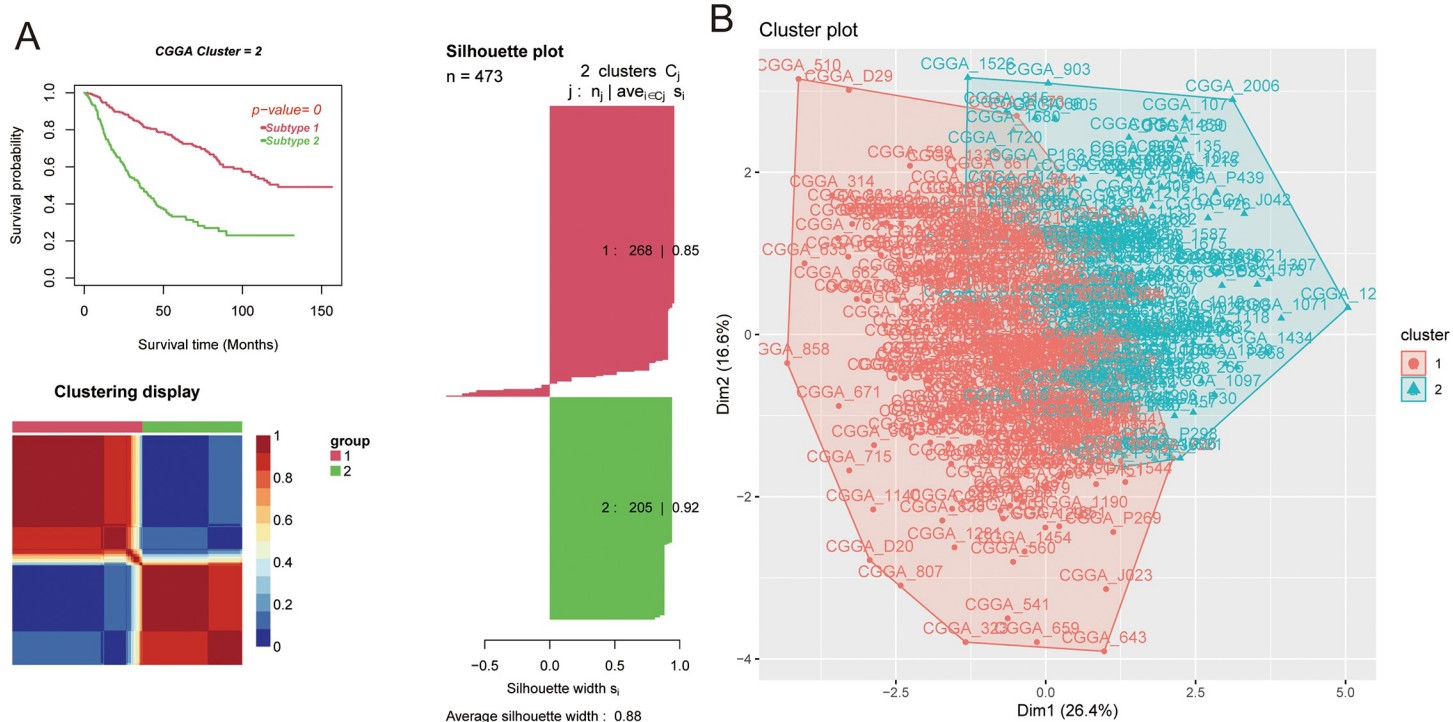

**Fig 2. Identification of RNA modification "writer" molecular subtypes.** (A) Kaplan–Meier survival analysis of two subtypes with log-rank test p value; clustering heatmap of two subtype samples; average silhouette width representing the coherence of clusters. (B) Principal component analysis shows that the two subtypes have significant differences in the transcriptome.

and C2) from the CGGA dataset were used for subsequent analyses. Kaplan-Meier curves showed that patients with the C1 subtype showed a longer OS than patients with the C2 subtype (Fig 2A; median OS, 10.07 [95% CI, 8.66-NA (not available)] vs 2.89 [95% CI, 2.29–3.62] years; log-rank test, $p < 0.001$). PCA showed differences between C1 and C2 (Fig 2B). In addition, by comparing clinicopathological characteristics between different subtypes, it was found that the expression of RNA modification "writer" genes significantly differed from the clinicopathological characteristics (S3A Fig). Compared with the C1 subtype, the C2 subtype showed a lower 1p19q co-deletion rate, lower *IDH* mutation rate, higher WHO grade, and higher risk of recurrence.

## Characteristics of the TME in different RNA modification subtypes

GSVA enrichment analysis was used to determine the biological significance of different RNA modification-based molecular subtypes (S3 Table in S2 File). GSVA enrichment analysis showed that C2 subtypes were significantly enriched for classic immune, carcinogenic, proliferation, and apoptosis signaling pathways, such as mismatch repair, primary immunodeficiency, EMC and receptor interaction, and the P53 signaling pathway (S3B Fig).

The CIBERSORT algorithm was used to evaluate the abundance of 22 types of human immune cells in LGG samples. Plasma cells, CD8 T cells, regulatory T cells (Tregs), and M1 macrophages showed a significantly lower degree of infiltration in the C1 subtype than in the C2 subtype, whereas naïve CD4 T cells, resting NK cells, and monocytes showed a higher infiltration degrees in the C1 subtype (Fig 3A). The estimate package was used to evaluate the stromal score, immune score, and tumor purity of LGG samples. Compared with the C1 subtype, the C2 subtype showed a higher stromal score and immune score (Fig 3B and 3C), and a lower

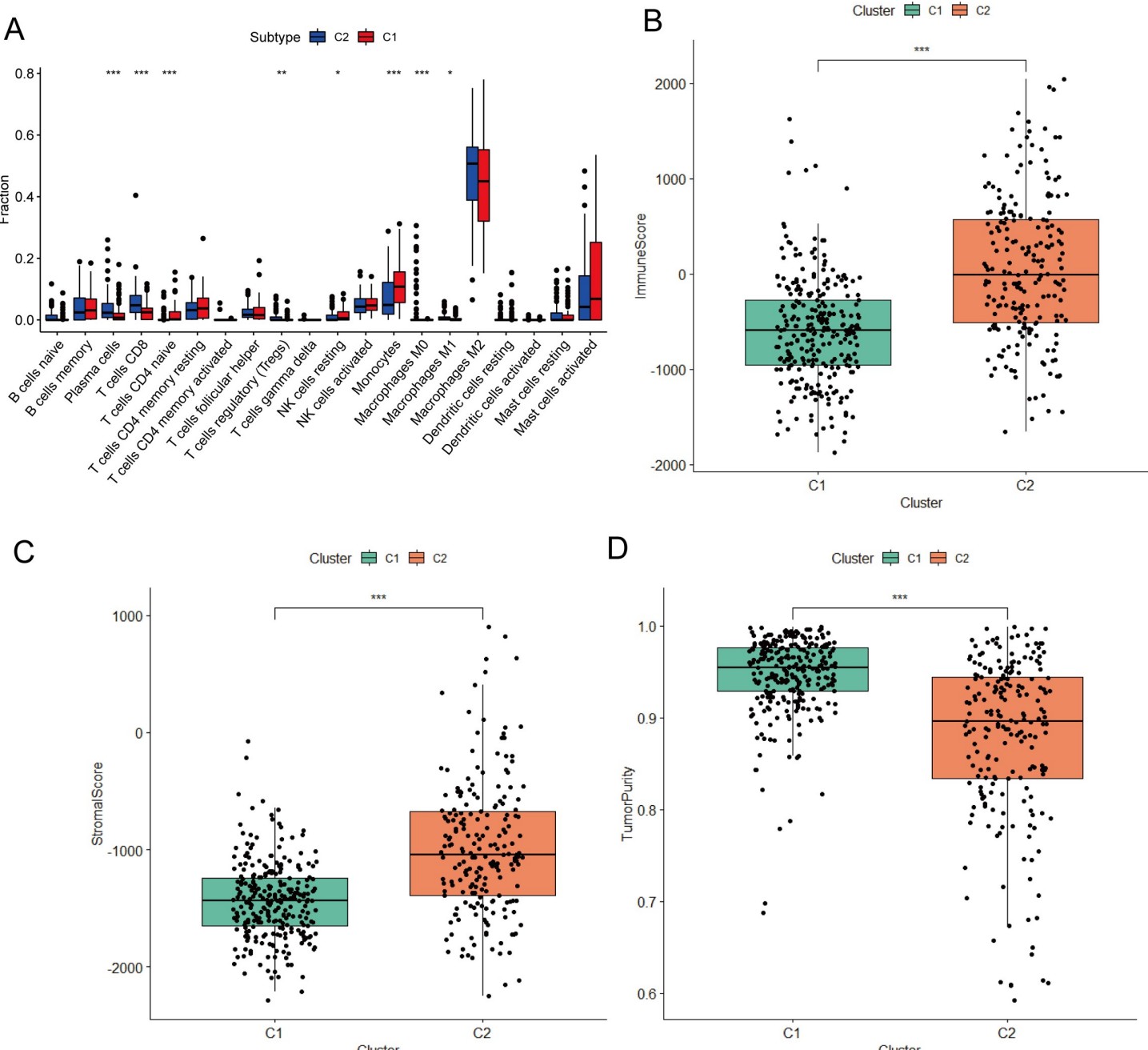

**Fig 3. Correlation analysis between RNA modification subtypes and immune microenvironment.** (A) Abundance of 22 infiltrating immune cell types in the two RNA modification subtypes. (B-D) Correlations between the two RNA modification subtypes and Immune Score (B), Stromal Score (C) and tumor purity (D).

tumor purity was lower (Fig 3D). These results suggested that RNA modification patterns may affect the remodeling of the immune microenvironment in LGG.

## Gene subtype identification based on DEGs

The R package "limma" was used to identify 616 differentially expressed genes (DEGs) related to the RNA modification-based tumor subtypes (S4 Table in S2 File), and subsequently

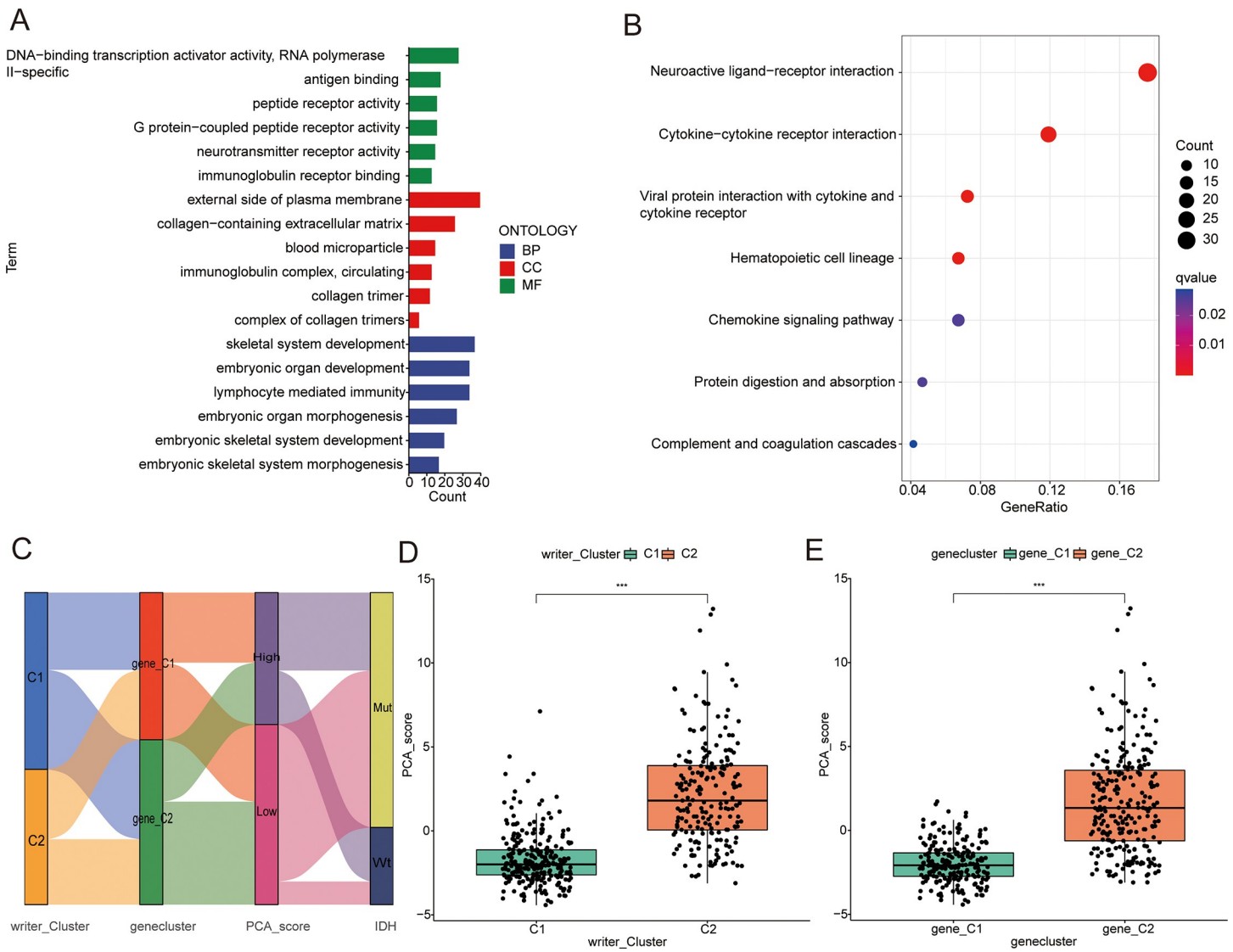

**Fig 4. Identification of gene subtypes based on DEGs.** (A and B) GO and KEGG enrichment analyses of DEGs among two RNA modification "writer" molecular subtypes. (C) Alluvial diagram of subtype distributions in groups with different PCA_score and IDH mutation. (D) Differences in PCA_score between two RNA modification "writer" molecular subtypes. (E) Differences in PCA_score between gene subtypes.

functional enrichment analysis was performed (S5 Table in S2 File). The DEGs were significantly enriched in biological processes related to immunity (Fig 4A). KEGG analysis showed that these DEGs were mainly enriched in neuroactive ligand-–receptor interactions, cytokine-–cytokine receptor interactions, and chemokine signaling pathways (Fig 4B).

In order to further verify these regulatory mechanisms, survival analysis was performed based on the expression of the 616 DEGs to determine their prognostic value (S6 Table in S2 File). Overall, 316 OS-related genes ($p < 0.05$) were selected for subsequent analysis. Based on the expression values of the prognosis-related genes, CNMF was used to divide samples from the CGGA-LGG cohort into two gene subtypes (ASW = 0.99): gene_C1 and gene_C2. Kaplan–Meier survival analysis showed that compared with the gene_C2 subtype, the gene_C1 subtype was associated with a significant survival advantage (S4A Fig). There was a significant

difference in the gene expression of RNA modification "writer" enzymes between the gene_C1 and gene_C2 groups (S4B Fig).

## Construction and verification of prognostic PCA scores

Due to the heterogeneity and complexity of RNA modifications, an RNA modification score model was developed to quantify the RNA modification patterns in individual LGG patients. DEGs were analyzed using the VNLHHO, and 16 characteristic genes were identified. Of these, the 15 genes that were also related to prognosis were selected to develop the PCA_score model. The distribution of LGG patients in the two RNA modification-based molecular subtype groups, gene subtype groups, PCA_score groups, and IDH mutation status groups was consistent with the expected values (Fig 4C). Compared with the C2 subtype, the C1 subtype showed a lower PCA_score (Fig 4D); in contrast, compared with the gene_C2 subtype, the gene_C1 subtype showed a higher PCA_score (Fig 4E). Further, there was a significant correlation between PCA_score and different clinical characteristics (S5 Fig).

The PCA_score risk distribution chart showed that as the PCA_score increased, the survival time decreased, and the number of deaths increased (Fig 5A). PCA analysis showed different dimensions in the High PCA_score group(H-PCA_score) and Low PCA_score group (L-PCA_score) (S6A Fig). Kaplan–Meier survival curves showed that the L-PCA_score had a better prognosis than the H-PCA_score (Fig 5B; median overall survival, 9.64[95%CI, 8.66-NA (not available)] vs 2.45[95%CI, 2.00–3.07] years; log-rank test, $p < 0.001$). In addition, the area under the curve (AUC) for the value of PCA_score in predicting 1-, 3-, 5-, and 7-year survival rates was 0.756, 0.777, 0.773, and 0.732, respectively (S6B Fig). The clinical characteristics of LGG patients were used to perform multivariate Cox analysis, which showed that PCA_score is a reliable and independent prognostic biomarker in LGG (Fig 5C; hazard ratio, 1.05 [95%CI, 1.01–1.09]; log-rank test, $p = 0.013$). The reliability of PCA_score was verified through an external validation dataset of LGG patients obtained from the TCGA database. In this cohort, Kaplan–Meier survival curves showed that patients with L-PCA_score had a better prognosis than patients with H-PCA_score (S7B Fig); the AUC of PCA_score in predicting 1-, 3-, 5-, and 7-year survival rates was 0.719, 0.739, 0.682, and 0.667, respectively (S7C Fig). Multivariate Cox analysis also revealed that patients with L-PCA_score had a significant survival advantage (S7D Fig). These results suggested that PCA_score was an independent prognostic factor for LGG, and had an excellent ability to predict the survival rate of LGG patients.

A nomogram was constructed based on the results of multivariate Cox analysis of data from the CGGA cohort (Fig 5D). The calibration chart of survival probability after 3, 5, and 7 years showed good consistency between the values predicted by the nomogram and actual values (Fig 5E). The C-index of PCA_score in predicting OS was higher than that of traditional markers (Fig 5F). The power of the nomogram and its individual factors in predicting OS was evaluated. The AUC values of the nomogram for predicting 3-, 5-, and 7-year survival were significantly higher than those of other factors (S7F Fig). DCA showed that the clinical application value of the prognostic model was better than that of other factors (S7G Fig). These results showed that the integration of PCA_score into the prediction model significantly improved its predictive power.

## Correlation of PCA_score with the TME

The relationship between PCA_score and the TME was evaluated based on Pearson correlation coefficients. With an increase in PCA_score, the immune score and stromal score also increased, while the tumor purity decreased (Fig 6A–6C). In addition, the PCA_score was significantly correlated with infiltration by various types of immune cells. For example, LGG

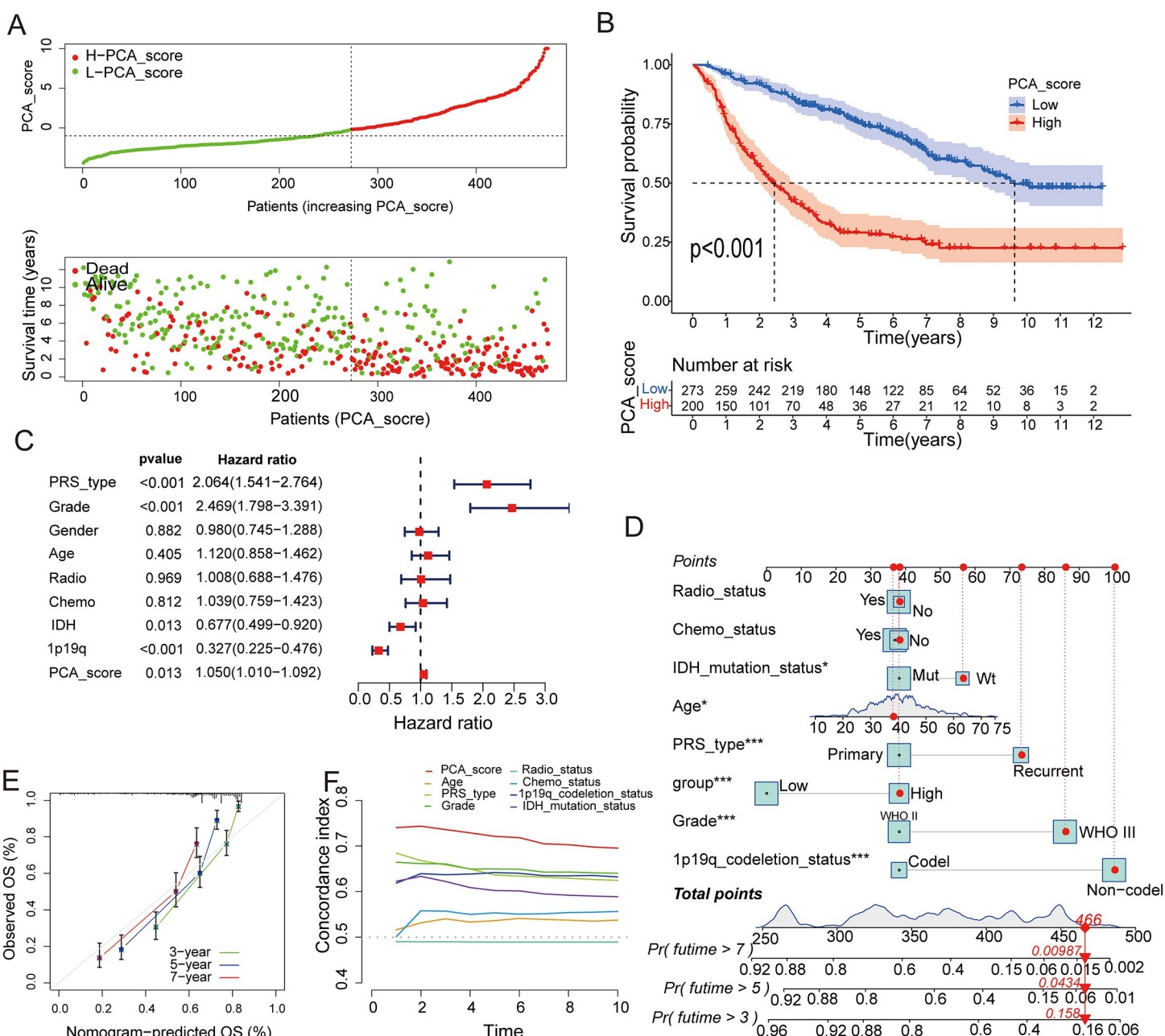

**Fig 5. Construction of the PCA_score in the training set.** (A) Ranked dot and scatter plots showing the PCA_score distribution and patient survival status. (B) Kaplan–Meier analysis of the OS between the two groups. (C) PCA_ Score and clinical characters were fitted for multivariate Cox analysis. (D) Nomogram for predicting the 3-, 5-, and 7-year OS of LGG patients in the training set. (E) Calibration curves of the nomogram for predicting of 3-,5-, and 7-year OS. (F) Concordance index to predict of survival according to the PCA_score.

patients with high PCA_score values, M0 macrophages, CD8 T cells, plasma cells, Tregs, M1 macrophages, follicular helper T cells, and naïve B cells showed a highly infiltration abundance; LGG patients with low PCA_score values, monocytes, CD4 T cells and resting NK cells showed a highly infiltration abundance (Fig 6D). The relationship between the 15 genes used to obtained PCA_score and the abundance of immune cells was also analyzed. The infiltration abundance of most immune cells was significantly correlated with the expression of these 15

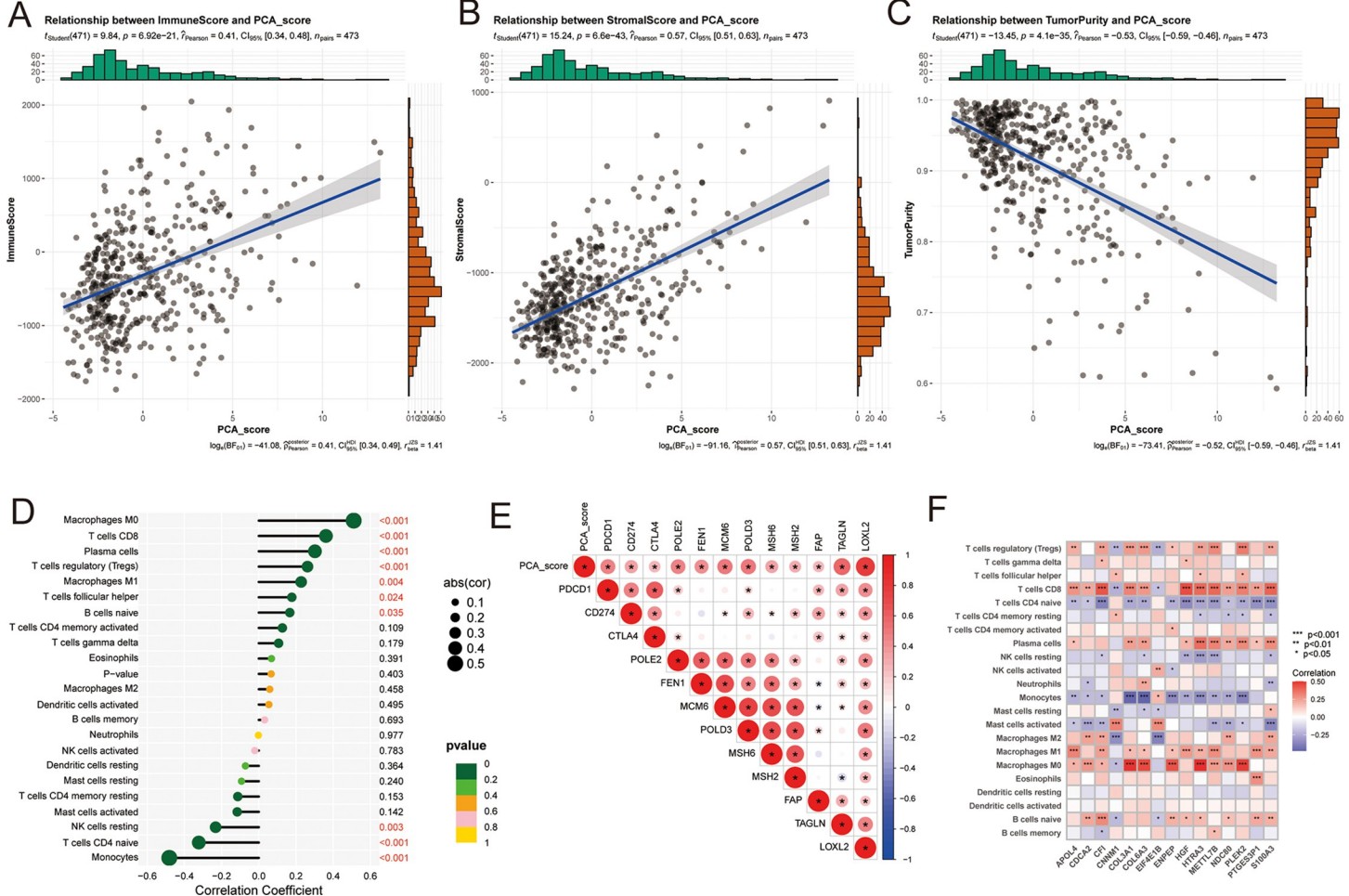

**Fig 6. Comprehensive analysis of PCA_score with TME in LGG patients.** Correlations between PCA_score and both immune score(A), stromal score(B) and tumor purity(C). (D) Correlations between PCA_score and immune cell infiltration abundance. (E) Correlation analysis between PCA_score and immune checkpoint genes. (F) Correlations between the abundance of immune cells and 15 genes in the proposed model.

genes (Fig 6E). In addition, analyses of the relationship between PCA_score and the expression of immune checkpoint genes (*PD-1*, *PD-L1*, *CTLA-4*, etc.) showed that with an increase in PCA_score, the expression of immune checkpoint genes also increased (Fig 6F). This suggested that PCA_score most likely affects the remodeling of immune infiltration in LGG and is a potential predictor of immunotherapy response in LGG patients.

## Correlation of PCA _score with mutations and drug sensitivity

It has been reported that patients with high TMB may be more sensitive to immunotherapy due to a higher number of neoantigens [37]. The relationship between TMB and PCA_score was analyzed in the TCGA-LGG cohort, and the TMB associated with H-PCA_score was found to be higher than that associated with L-PCA_score (Fig 7A). Pearson correlation analysis revealed that there was a significant positive correlation between PCA_score and TMB (Fig 7B), suggesting that LGG patients with a high PCA_score may benefit from immunotherapy. Then, somatic mutations were analyzed in the H-PCA_score and L-PCA_score TCGA-LGG cohorts. Overall, 25 and 7 high-frequency mutation genes were detected in the H-PCA _score and L-PCA_score cohorts, respectively (Fig 7C and 7D). Compared with H-PCA_score,

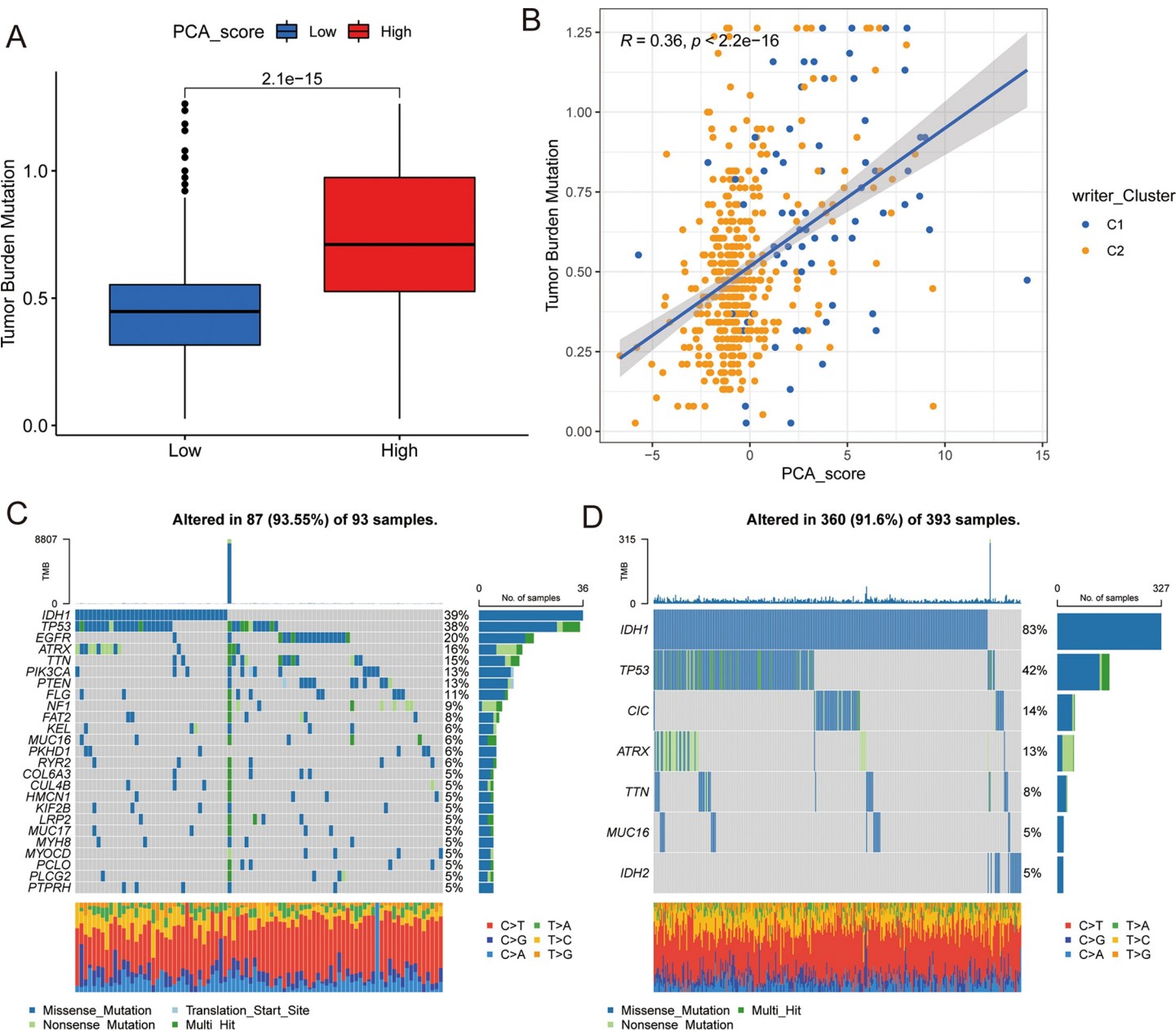

**Fig 7. Correlation analysis of PCA_score and TCGA-LGG cohort somatic mutations.** (A) TMB in different PCA_score groups. (B) pearson correlation analysis of the PCA_score and TMB. (C and D) The waterfall plot of somatic mutation features established with high and low PCA_scores. Each column represented an individual patient. The upper barplot showed TMB, the number on the right indicated the mutation frequency in each gene. The right barplot showed the proportion of each variant type.

L-PCA_score was associated with a higher mutation frequency for *IDH1* and *CIC* and a lower mutation frequency for *EGFR* and *PIK3CA* (S7 Table in S2 File). In addition, the IC50 values axitinib and gefitinib were higher for H-PCA_score than for L-PCA_score, and the IC50 values of bexarotene, rapamycin, vinblastine, gemcitabine, docetaxel, and metformin were lower (S8 Fig). Together, the findings indicated that PCA_score could be a predictive marker of the response to the aforementioned therapeutic agents.

### Influence of PCA_score on the outcomes of adjuvant therapy

Kaplan-Meier survival analysis was performed on the Chemo_group and the non-Chemo_group from the CGGA-LGG cohort. Compared with LGG patients in the Chemo_group, those in the non-Chemo_group had a better prognosis (S9A Fig). The prognosis of LGG patients in the Chemo_group and the non-Chemo_group was further analyzed based on PCA_score. The prognosis of LGG patients in the Chemo_group with a low PCA_score was worse than that of those in the non-Chemo_group (S9B Fig), while there was no such difference among those with a high PCA_score value (S9C Fig). Among LGG patients with low PCA_score values, those in the non-Chemo_group undergoing radiotherapy had the best prognosis (S9D Fig). Among LGG patients with high PCA_score values, chemotherapy combined with radiotherapy had no significant effect on prognosis (S9E Fig).

The correlation between PCA_score and the TME may be suggestive of an environment that promotes cell rejection and dysfunction. Based on this, the TIDE algorithm was used to assess the level of T cell dysfunction and rejection. In LGG, PCA_score was negatively correlated with the TIDE score (Fig 8A). Compared with L-PCA_score, H-PCA_score was associated with a higher proportion of LGG patients who responded to immunotherapy (Fig 8B). The PCA_score of the "TRUE" response group to immunotherapy was higher than that of the "FALSE" response group (Fig 8C). This indicated that these patients with H-PCA_score had lower levels of immune dysfunction and better responses to immunotherapy.

Immunotherapy through *PD-L1* and *PD-1* blockade has been a major breakthrough in cancer treatment. Using the IMvigor210 cohort, the effect of RNA modification characteristics on the response to immune checkpoint lockout therapy was examined. The prognostic value of PCA_score was examined in the TCGA-BLCA cohort, and H-PCA-score was significantly associated with low survival (Fig 8D; median OS, 7.24[95%CI, 2.75-NA (not available)] vs 1.97 [95%CI, 1.65–2.67] years; log-rank test, $p < 0.001$). In the IMvigor210 cohort, patients with H-PCA_score values showed a significantly prolonged survival (Fig 8E; medianOS, 16.26[95% CI, 10.18-NA (not available)] vs 7.95[95%CI, 6.70–9.89] years; log-rank test, $p < 0.001$). It was confirmed that compared with L-PCA_score patients, H-PCA_score patients showed significant therapeutic advantages and treatment efficacy after immunotherapy (Fig 8F and 8G).

In summary, the research showed that RNA modification patterns are significantly related to the response to LGG immunotherapy, and the established RNA modification scoring model could help in predicting the response to immunotherapy.

## Discussion

In this study, the overall changes in the transcriptional and genetic levels of RNA modifications in LGG were examined. Based on 26 RNA modification "writer" enzymes, two different LGG molecular subtypes were identified, and the OS of these different "writer" molecular subtypes was found to be significantly different. The characteristics of the TME and the degree of enrichment in classical immune, carcinogenic, proliferation and apoptosis signaling pathways were also significantly different between the two subtypes. The DEGs between the two "writer" molecular subtypes were found to be involved in immune-related biological pathways. VNLHHO, a new meta-heuristic method for gene feature extraction that balances global exploration and local development, was used to identify the characteristic genes regulated by RNA modification in LGG. Here, DEGs were analyzed according to VNLHHO to obtain characteristic genes and construct a prognostic PCA_score model in order to evaluate the effect of the RNA modification "writers" in individual patients. Compared with LGG patients showing low PCA_score values, those with high PCA_score values showed a worse prognosis. Finally, by combining PCA_score with results of multivariate Cox analysis results, a nomogram was

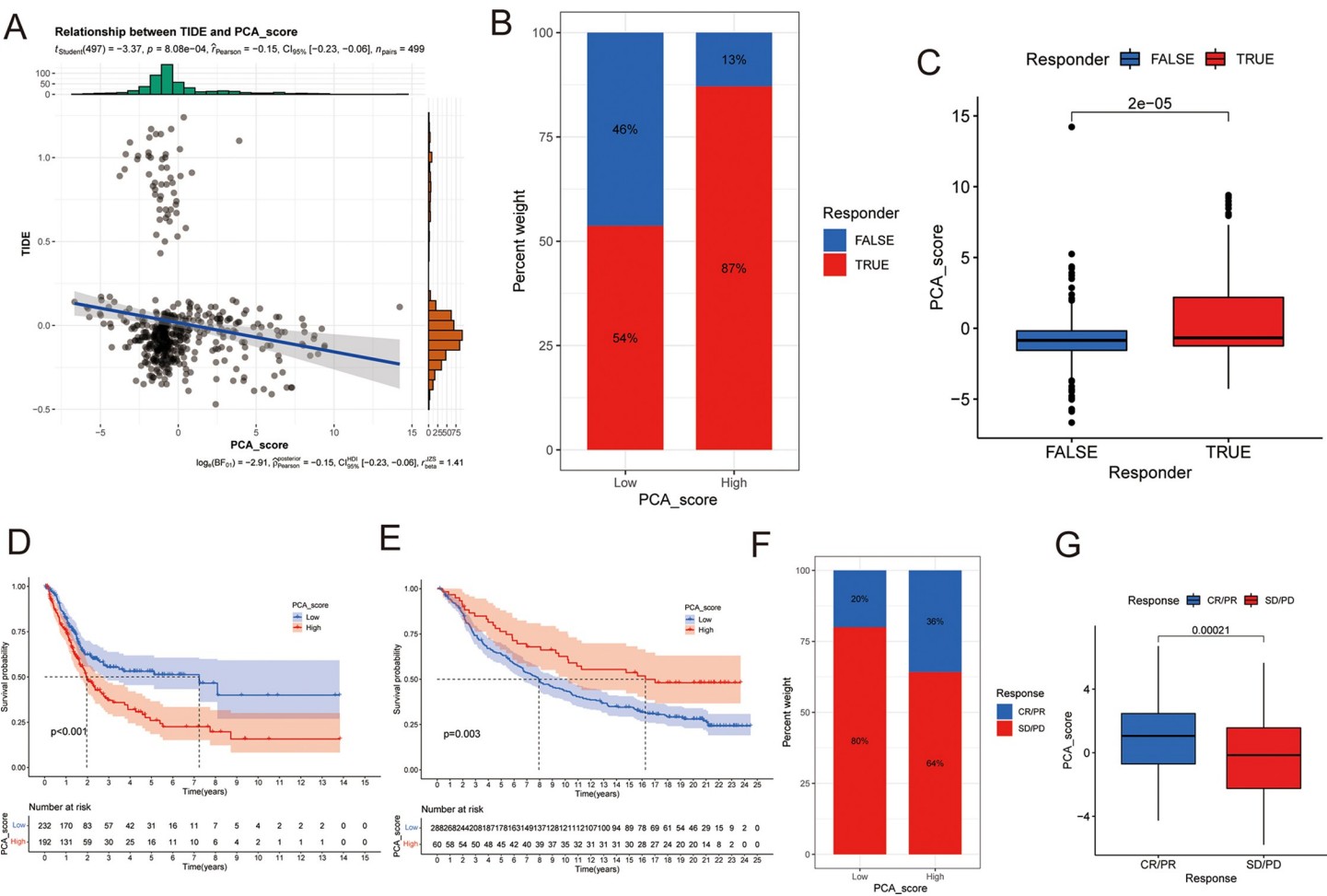

**Fig 8. Comprehensive analysis of PCA_score with immunotherapy response in LGG patients.** (A) Correlations between PCA_score and TIDE. (B) Proportion of LGG patients with H-PCA_score and L-PCA_score responding to immunotherapy. (C) Difference of PCA_score between "TURE" group and "FLASE" group. (D) Survival analyses for L-PCA_score and H-PCA_score patient in TCGA-BLCA cohort using Kaplan-Meier curves. (E) Survival analyses for L-PCA_score and H-PCA_score patient in IMvigor210 cohort using Kaplan-Meier curves. (F) The proportion of patients with response to PD-L1 blockade immunotherapy in L-PCA_score and H-PCA_score. (G) Distribution of PCA_score in distinct anti-PD-L1 clinical response groups.

established. This nomogram further improved the predictive value of PCA_score. The prognostic model can improve our understanding of differences in molecular mechanisms between individuals with LGG and provide new insights for treatment strategies such as precision therapy.

The conventional treatment plan for LGG is still radiotherapy, chemotherapy, and surgery. However, the results of this study showed that chemotherapy is not effective in treating LGG, and it may even lead to a worse prognosis. The TME is known to be associated with tumor progression and treatment resistance [38]. Although gliomas are considered "cold tumors" with few infiltrating immune cells [39], in gliomas, Treg infiltration is associated with IDH mutations and helps determine the best response to treatment [40]. Mast cells participate in the promotion of tumor angiogenesis and can affect the tumor's anti-immune response [41, 42]. The C1 subtype is associated with low PCA_score and naïve CD4 T cell infiltration, resting NK cell infiltration, and high monocyte infiltration, suggesting that these cells play an active role in the development of LGG. The C2 subtype is significantly correlated with high PCA_-score. Tregs, plasma cells, CD8 T cells, and M0 and M1 macrophages show a high degree of infiltration in this subtype. It has been reported that CD8 T cell infiltration is negatively

correlated with glioma progression [43]. Glioma-associated macrophages promote tumor proliferation, invasion, metastasis, and angiogenesis [44, 45]. Therefore, improved understanding of the immune microenvironment of gliomas could promote the development of new precision treatments for these tumors.

The expansion of tumor immunology research, development of molecular biology technology, and emergence of immunotherapy, including immune checkpoint inhibitors (ICIs), have provided novel strategies for tumor treatment. Research on ICIs against *CTLA-4*, *PD-1*, and *PD-L1* is booming, and clinical studies have confirmed their safety and effectiveness [46, 47]. This study showed that *PD-1* and *PD-L1* levels are positively correlated with PCA_score, but there is no significant correlation with the TMB. A significant correlation was also found to exist between PCA_score and the TIDE score in predicting the clinical response of LGG patients to immune checkpoint blockade. Therefore, LGG patients with high PCA_score values may respond to immunotherapy.

Although our comprehensive analysis of LGG genomic data from independent cohorts yielded valuable conclusions with potential for clinical translation, this study also has certain limitations. For example, all analyses were performed using the public databases CGGA and TCGA, and all samples were obtained retrospectively. Therefore, large-scale prospective studies and additional in vivo and in vitro experimental studies are needed to validate our findings. For example, detect the expression of these newly discovered glioma prognosis related genes in clinical samples and analyze their relationship with the prognosis of glioma patients and the infiltration status of immune cells, and analyze the impact of these genes on the biological functions of glioma cells such as proliferation and invasion in vitro; And detect the influence of these genes on immune function in vivo through gene overexpression/knockout technology.

Overall, this comprehensive analysis of RNA modification "writer" genes reveals their broad regulatory mechanisms that are linked to LGG prognosis and the immune microenvironment. We constructed a PCA_score prognostic model to quantify RNA modification patterns in individual LGG patients and identified the potential of PCA_score in guiding targeted therapy and immunotherapy. These findings highlight the clinical significance of RNA modifications in LGG and pave the way for personalized immunotherapy strategies for LGG patients.

## Supporting information

**S1 File.**
(DOCX)

**S2 File.**
(XLSX)

**S1 Data.**
(R)

**S1 Fig. Expression distribution of 26 RNA modification "Writer" enzyme genes in normal tissues and LGG tissues.**
(TIF)

**S2 Fig. Identification of RNA modification subtypes in the TCGA-LGG cohort.**
(TIF)

**S3 Fig.** Differences in clinical characteristics (A) and biological processes (B) between the two RNA modification subtypes.
(TIF)

**S4 Fig. Gene subtypes identification of characteristic genes based on RNA modification.**
(TIF)

**S5 Fig. Differences of PCA_score in different clinical traits.**
(TIF)

**S6 Fig. PCA analysis and ROC curve analysis.**
(TIF)

**S7 Fig. Validation of the PCA_score prognostic model of the TCGA-LGG cohort.**
(TIF)

**S8 Fig. Correlation between PCA_score and IC50 of common Chemotherapy drugs.**
(TIF)

**S9 Fig. Kaplan–Meier survival curve of Radio group and non-Chemo group in H-PCA_-score.**
(TIF)

## Acknowledgments

We acknowledge TCGA database and CGGA database for providing a platform and contributors for uploading their meaningful datasets.

## Author Contributions

**Conceptualization:** Lupeng Zhang, Xiaoning Peng.

**Data curation:** Lupeng Zhang, Chen Shi, Yifan Tang.

**Formal analysis:** Lupeng Zhang, Chiwen Qu, Chen Shi, Suye Zhong.

**Funding acquisition:** Xiaoning Peng.

**Methodology:** Chiwen Qu, Fan Wu, Yifan Tang, Jinlong Li.

**Project administration:** Xiaomin Zeng, Xiaoning Peng.

**Writing – original draft:** Lupeng Zhang, Chen Shi, Huicong Feng, Jun Yang.

**Writing – review & editing:** Lupeng Zhang, Yue Li, Xiaomin Zeng, Xiaoning Peng.

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
