## [Decision Letter · Decision Letter 0]

14 Jul 2022

PONE-D-22-11089Association of RNA-modification "writer" genes with prognosis and response to immunotherapy in patients with low-grade gliomaPLOS ONE

Dear Dr. Peng,

Thank you for submitting your manuscript to PLOS ONE. After careful consideration, we feel that it has merit but does not fully meet PLOS ONE’s publication criteria as it currently stands. Therefore, we invite you to submit a revised version of the manuscript that addresses the points raised during the review process.

We look forward to receiving your revised manuscript.

Kind regards,

Girijesh Kumar Patel, PhD

Academic Editor

PLOS ONE

Journal Requirements:

“This project was supported by the National Natural Science Foundation of China (grant number 81472860, 61761001), Key R & D project of Hunan Province (grant number 2020DK2002) and the research start-up fund for Prof. Peng Xiaoning from Jishou University (grant number 91602-111900).”

Reviewers' comments:

Reviewer's Responses to Questions

**Comments to the Author**

1. Is the manuscript technically sound, and do the data support the conclusions?

Reviewer #1: Yes

Reviewer #2: No

2. Has the statistical analysis been performed appropriately and rigorously? 

Reviewer #1: Yes

Reviewer #2: N/A

3. Have the authors made all data underlying the findings in their manuscript fully available?

Reviewer #1: No

Reviewer #2: Yes

4. Is the manuscript presented in an intelligible fashion and written in standard English?

Reviewer #1: Yes

Reviewer #2: No

5. Review Comments to the Author

Reviewer #1: The main concern for this paper from the technical point of view is that the included data from TCGA and CGGA databases are not included in the supplementary materials.

Moreover, for reproducibility purposes, readers need to access the R codes used in this research. Many R packages were mentioned in the paper to conduct the analysis, such as

"CancerSubtypes, "SVA", "CNMF", "survival", "limma", "cluster profile", "rms", "maftools", “pRRophetic”, "IMvigor210CoreBiologies"," DEseq2", but I was not able to find any of them in the supplementary materials either.

Other than these two issues, the logic behind the analysis and explanation of each one seems reasonable.

Reviewer #2: Dear Authors

First, I want to compliment and thank you all you all for exploring such a complex subject.

The aim of the article is to explore the alterations in the expression and mutations of RNA-modification writer genes and to establish the connection of these modifications with prognosis and response to chemotherapy in patients with low grade glioma. The methods to investigate the problem are satisfactory, however following are the suggestion to improve the quality of the manuscript.

1-The language of the manuscript is extremely ambiguous and not scientific.

I suggest the authors to restructure and rewrite the abstract clear the message as well as to make the language more scientific, less complex and free of most grammatical and typographical errors.

2-A lot of abbreviations have been used but there is no table or figure for the full forms. The abbreviated words have not been described even at the first place.

3- The scientific background and rationale of the investigation, and the methodology in the manuscript is satisfactory but needs more description. I would suggest the authors to provide the clear aim, hypothesis and objectives of the study.

4-The analysis of the results and the Bioinformatics as well as statistical methods applied in the manuscript are satisfactory and all the data presented in the manuscript is descriptive, and the results are presented in the forms of figures. However, almost 99% of the figures need to be restructured, and the legends should be readable (figure 1B, 1D-nothing is visible, 2A, 2B, 3B, C D, 4A, B, C, D, 5C, D, E, F, G, H, 6 A, B, C, D, E, F). There is repetition of some figures.

5-All the key results should be summarized with reference to study objectives.

6-The interpretation of some of the results is in the light of original results but the limitations of the study have not been discussed. There are several places where references have not been provided.

7-There is no experimental validation which can support the findings of the study.

7-The most important and major concern is the inclusion of urothelial cancer data in the manuscript. The manuscript is about low-grade glioma so why the authors have provided the methodology (page number 9 and 10) and figures (Figure 8A, B, C, D, E) about urothelial cancer that too without any justification or co-relation.

I, would recommend the authors to restructure and re-write the article and remove all the data from other studies. After the inclusion of all the correction, the manuscript should be sent to an expert of English language.

6. PLOS authors have the option to publish the peer review history of their article (what does this mean?). If published, this will include your full peer review and any attached files.

Reviewer #1: No

Reviewer #2: **Yes: **Sabiha khatoon

---

## [Author Response · Author response to Decision Letter 0]

4 Aug 2022

Reply to Reviewer 1

Point 1: The main concern for this paper from the technical point of view is that the included data from TCGA and CGGA databases are not included in the supplementary materials. Moreover, for reproducibility purposes, readers need to access the R codes used in this research. Many R packages were mentioned in the paper to conduct the analysis, such as "CancerSubtypes, "SVA", "CNMF", "survival", "limma", "cluster profile", "rms", "maftools", “pRRophetic”, "IMvigor210CoreBiologies"," DEseq2", but I was not able to find any of them in the supplementary materials either.

Other than these two issues, the logic behind the analysis and explanation of each one seems reasonable.

Response : Thanks for the insightful suggestion. Based on your comments, we have supplemented the corresponding data and codes to make our results more convincing. The code for the key steps used in this paper has been compiled and added to the Supplementary Material. Also, since the data we used came from publicly available databases, they were not uploaded. These data can be found freely at TCGA (https://portal.gdc.cancer.gov/ ) and CGGA (http://www.cgga.org.cn/).

Reply to Reviewer 2

Point 1: The language of the manuscript is extremely ambiguous and not scientific.

I suggest the authors to restructure and rewrite the abstract clear the message as well as to make the language more scientific, less complex and free of most grammatical and typographical errors.

Response : Thank you very much for your criticism. To improve the quality of the manuscript, we have made extensive revisions to the manuscript and marked all revisions in the text. We have rewritten the Abstract section and carefully proofread the manuscript to minimize typographical and grammatical errors.

Point 2: A lot of abbreviations have been used but there is no table or figure for the full forms. The abbreviated words have not been described even at the first place.

Response : Thanks for your detailed suggestions. We reviewed the full text thoroughly, added a list of abbreviations before the abstract, and described abbreviations in detail when they first appeared.

Point 3: The scientific background and rationale of the investigation, and the methodology in the manuscript is satisfactory but needs more description. I would suggest the authors to provide the clear aim, hypothesis and objectives of the study.

Response : Thank you very much for your professional advice. We have described our background in detail and provided clear research aim and hypothesis.

Point 4: The analysis of the results and the Bioinformatics as well as statistical methods applied in the manuscript are satisfactory and all the data presented in the manuscript is descriptive, and the results are presented in the forms of figures. However, almost 99% of the figures need to be restructured, and the legends should be readable (figure 1B, 1D-nothing is visible, 2A, 2B, 3B, C D, 4A, B, C, D, 5C, D, E, F, G, H, 6 A, B, C, D, E, F). There is repetition of some figures.

Response : Thanks a lot. We have reviewed all the Figures in the text, reordered the numbers, and adjusted the legend section to make it readable.

Point 5: All the key results should be summarized with reference to study objectives.

Response : Thank you very much. We have summarized the main findings with reference to the research objectives and annotated them in the text.

Point 6: The interpretation of some of the results is in the light of original results but the limitations of the study have not been discussed. There are several places where references have not been provided.

Response : Thanks for your professional review. we have elaborated on the limitations of this study in the Discussion. In addition, corresponding supplements were made where references were lacking. details as follows:

1. We cited references in "Analysis of the correlation between LGG molecular subtypes and the TME" in Methods: Yoshihara K, Shahmoradgoli M, Martínez E, Vegesna R, Kim H, Torres-Garcia W, et al. Inferring tumour purity and stromal and immune cell admixture from expression data. Nature communications. 2013;4:2612. doi: 10.1038/ncomms3612. PubMed PMID: 24113773.

2. We cited references in Methods "Differentially expressed gene (DEG) identification, functional annotation and VNLHHO": Ritchie M, Phipson B, Wu D, Hu Y, Law C, Shi W, et al. limma powers differential expression analyses for RNA-sequencing and microarray studies. Nucleic acids research. 2015;43(7):e47. doi: 10.1093/nar/gkv007. PubMed PMID: 25605792.

3. We cited references in the method "Analysis of the correlation of PCA_score with LGG somatic mutations and drug sensitivity": Geeleher P, Cox N, Huang R. pRRophetic: an R package for prediction of clinical chemotherapeutic response from tumor gene expression levels. PloS one. 2014;9(9):e107468. doi: 10.1371/journal.pone.0107468. PubMed PMID: 25229481.

Point 7: There is no experimental validation which can support the findings of the study.

Response : This issue is one of the limitations of this study. We will further validate our findings in future work so that the findings are more convincing.

Point 8: The most important and major concern is the inclusion of urothelial cancer data in the manuscript. The manuscript is about low-grade glioma so why the authors have provided the methodology (page number 9 and 10) and figures (Figure 8A, B, C, D, E) about urothelial cancer that too without any justification or co-relation.

Response : In the present study, we found that PCA_score may be a potential predictor of response to LGG immunotherapy, but no immunotherapy-treated LGG cohort could be used for confirmation. Therefore, we refer to the following articles: 

1. Chang P, Chen S, Chang X, Zhu J, Tang Q, Ma L. EXTL3 could serve as a potential biomarker of prognosis and immunotherapy for prostate cancer and its potential mechanisms. European journal of medical research. 2022;27(1):115. doi: 10.1186/s40001-022-00740-w. PubMed PMID: 35818069

2. Peng S, Wang R, Zhou Y, Wei W, Zhong G, Huang X, et al. Insight of a Metabolic Prognostic Model to Identify Tumor Environment and Drug Vulnerability for Lung Adenocarcinoma. Frontiers in immunology. 2022;13:872910. doi: 10.3389/fimmu.2022.872910. PubMed PMID: 35812404

3. Song P, Li W, Guo L, Ying J, Gao S, He J. Identification and Validation of a Novel Signature Based on NK Cell Marker Genes to Predict Prognosis and Immunotherapy Response in Lung Adenocarcinoma by Integrated Analysis of Single-Cell and Bulk RNA-Sequencing. Frontiers in immunology. 2022;13:850745. doi: 10.3389/fimmu.2022.850745. PubMed PMID: 35757748.

4. Zhang H, Deng S, Pi Y, Guo J, Xi H, Shi X, et al. Identification and Validation in a Novel Quantification System of Ferroptosis Patterns for the Prediction of Prognosis and Immunotherapy Response in Left- and Right-Sided Colon Cancer. Frontiers in immunology. 2022;13:855849. doi: 10.3389/fimmu.2022.855849. PubMed PMID: 35444656.

These authors identified potential predictors of immunotherapy response in different cancers, and then used the IMvigor210 cohort to validate the predictors for immunotherapy response.

Point 9: I would recommend the authors to restructure and re-write the article and remove all the data from other studies. After the inclusion of all the correction, the manuscript should be sent to an expert of English language.

Response : According to your suggestion, we have rewritten this manuscript. For example, we added a list of abbreviations and rewrote the summary section. We rewrote the introduction to describe the background in more depth and clarify the purpose of this study. We rewrote the discussion part of this manuscript, discussed the limitations of this study, and summarized the results of this study. Finally, we polished the manuscript in a professional English editing company to correct English language, grammar, punctuation and phrasing to improve the clarity and readability of academic papers.

---

## [Decision Letter · Decision Letter 1]

11 Oct 2022

PONE-D-22-11089R1Association of RNA-modification "writer" genes with prognosis and response to immunotherapy in patients with low-grade gliomaPLOS ONE

Dear Dr. Peng,

Thank you for submitting your manuscript to PLOS ONE. After careful consideration, we feel that it has merit but does not fully meet PLOS ONE’s publication criteria as it currently stands. Therefore, we invite you to submit a revised version of the manuscript that addresses the points raised during the review process.

We look forward to receiving your revised manuscript.

Kind regards,

Girijesh Kumar Patel, PhD

Academic Editor

PLOS ONE

Journal Requirements:

Reviewers' comments:

Reviewer's Responses to Questions

**Comments to the Author**

1. If the authors have adequately addressed your comments raised in a previous round of review and you feel that this manuscript is now acceptable for publication, you may indicate that here to bypass the “Comments to the Author” section, enter your conflict of interest statement in the “Confidential to Editor” section, and submit your "Accept" recommendation.

Reviewer #2: All comments have been addressed

Reviewer #3: All comments have been addressed

2. Is the manuscript technically sound, and do the data support the conclusions?

Reviewer #2: Yes

Reviewer #3: Yes

3. Has the statistical analysis been performed appropriately and rigorously? 

Reviewer #2: I Don't Know

Reviewer #3: Yes

4. Have the authors made all data underlying the findings in their manuscript fully available?

Reviewer #2: Yes

Reviewer #3: Yes

5. Is the manuscript presented in an intelligible fashion and written in standard English?

Reviewer #2: Yes

Reviewer #3: Yes

6. Review Comments to the Author

Reviewer #2: Dear Authors,

Thank you for all the corrections.

My only suggestion is to improve the quality of figures. The legends in most of the figures are still not readable.

Thank you

Reviewer #3: In their article entitled “ "Association of RNA-modification "writer" genes with prognosis and response to immunotherapy in patients with low-grade glioma," Zhang et al. aimed to present a novel RNA modification-based prognostic model for LGG that would serve as the basis for assessing LGG prognosis and creating more potent therapeutic approaches for the LGG.

This constitutes a large, comprehensive, and broadly rational body of work that is appreciable. However, there are a number of minor concerns that can be resolved to improve the quality of the manuscript, listed below.

1. In the line No. 75 the author stated that “due to disease heterogeneity and drug resistance, the prognosis of LGG has not improved significantly”. Please elaborate on the relationship between drug resistance and the prognosis for LGG.

2. The figures are difficult to see and consequently understand, therefore please make extra effort to improve their quality and clarity.

3. The reasoning behind placing the figure legends in the center of the results section. Please create a separate section for all of the combined figure legends.

4. Please explain how these results relate to the validity of the in-vitro, in-vivo implementations.

5. The authors accidentally added their previously submitted revised version (with track change) (from page 60). When submitting to a reputable publication like Plos one, writers are kindly asked to pay more attention to prevent this sort of neglect.

7. PLOS authors have the option to publish the peer review history of their article (what does this mean?). If published, this will include your full peer review and any attached files.

Reviewer #2: **Yes: **sabiha khatoon

Reviewer #3: **Yes: **Susmita Barman

---

## [Author Response · Author response to Decision Letter 1]

17 Oct 2022

Response to Reviewer

Point 1: In the line No. 75 the author stated that “due to disease heterogeneity and drug resistance, the prognosis of LGG has not improved significantly”. Please elaborate on the relationship between drug resistance and the prognosis for LGG.

Response: Thank you very much for your professional advice. In the second paragraph of the introduction, we have described relationship between drug resistance and the prognosis for LGG in detail and added a new reference for this issue（Wang J, Cazzato E, Ladewig E, Frattini V, Rosenbloom D, Zairis S, et al. Clonal evolution of glioblastoma under therapy. Nature genetics. 2016;48(7):768-76. doi: 10.1038/ng.3590. PubMed PMID: 27270107.）

Point 2: The figures are difficult to see and consequently understand, therefore please make extra effort to improve their quality and clarity.

Response: Thank you very much for your professional advice. We have remade Figure 1, Figure 2, Figure 3, Figure 4, Figure 5, and improved their clarity. We have renamed the previous version of Figures 5B and 5D as Figure S6. The previous Figure S6, Figure S7, and Figure S8 were changed to Figure S7, Figure S8, and Figure S9 in turn.

Point 3: The reasoning behind placing the figure legends in the center of the results section. Please create a separate section for all of the combined figure legends.

Response: That’s ture. We added a separate section for all of the combined figure legends.

Point 4: Please explain how these results relate to the validity of the in-vitro, in-vivo implementations.

Response: Thanks a lot. In our discussion, we added a description of how to use in vivo and in vitro experiments to confirm the validity of our research results.

Point 5: The authors accidentally added their previously submitted revised version (with track change) (from page 60). When submitting to a reputable publication like Plos one, writers are kindly asked to pay more attention to prevent this sort of neglect.

Response: Thanks a lot. We have correctly uploaded the second revised version of our manuscript.

---

## [Decision Letter · Decision Letter 2]

1 Dec 2022

Association of RNA-modification "writer" genes with prognosis and response to immunotherapy in patients with low-grade glioma

PONE-D-22-11089R2

Dear Dr. Peng,

We’re pleased to inform you that your manuscript has been judged scientifically suitable for publication and will be formally accepted for publication once it meets all outstanding technical requirements. Moreover, based on the one reviewers it should be edited/modified accordingly.

Kind regards,

Girijesh Kumar Patel, PhD

Academic Editor

PLOS ONE

Additional Editor Comments (optional):

Dear Authors,

Based on the authors comments and my own evaluation. The manuscript is at the stage to be accepted. However, based on the suggestion by one reviewers it need to be modified.

Reviewers' comments:

Reviewer's Responses to Questions

**Comments to the Author**

1. If the authors have adequately addressed your comments raised in a previous round of review and you feel that this manuscript is now acceptable for publication, you may indicate that here to bypass the “Comments to the Author” section, enter your conflict of interest statement in the “Confidential to Editor” section, and submit your "Accept" recommendation.

Reviewer #2: All comments have been addressed

Reviewer #3: All comments have been addressed

2. Is the manuscript technically sound, and do the data support the conclusions?

Reviewer #2: Yes

Reviewer #3: Yes

3. Has the statistical analysis been performed appropriately and rigorously? 

Reviewer #2: Yes

Reviewer #3: Yes

4. Have the authors made all data underlying the findings in their manuscript fully available?

Reviewer #2: Yes

Reviewer #3: Yes

5. Is the manuscript presented in an intelligible fashion and written in standard English?

Reviewer #2: Yes

Reviewer #3: Yes

6. Review Comments to the Author

Reviewer #2: Dear Authors,

Thank you for implementing all the correction.

The manuscript is scientifically sound but again I have few concerns about about figures.

1-The arrangement of figures is not correct (figure 7, 8, 1, 3, 6, 5, 4,2).

2-There is nothing visible in figure 2B. It should be reconstructed or you may transfer it in supplementary information.

Thank you

Reviewer #3: In their article entitled "Association of RNA-modification "writer" genes with prognosis and response to immunotherapy in patients with low-grade glioma," Zhang et al. aimed to present a novel RNA modification-based prognostic model for LGG that would serve as the basis for assessing LGG prognosis and creating more potent therapeutic approaches for the LGG.

This constitutes a large, comprehensive, and broadly rational body of work that is appreciable.

The authors have addressed all the comments.

7. PLOS authors have the option to publish the peer review history of their article (what does this mean?). If published, this will include your full peer review and any attached files.

Reviewer #2: **Yes: **sabiha khatoon

Reviewer #3: **Yes: **Susmita Barman

---

## [Editor Report · Acceptance letter]

4 Jan 2023

PONE-D-22-11089R2 

Association of RNA-modification “writer” genes with prognosis and response to immunotherapy in patients with low-grade glioma 

Dear Dr. Peng:

I'm pleased to inform you that your manuscript has been deemed suitable for publication in PLOS ONE. Congratulations! Your manuscript is now with our production department. 

Kind regards, 

on behalf of

Dr. Girijesh Kumar Patel 

Academic Editor

PLOS ONE